# Biosurfactant Production and Growth Kinetics Studies of the Waste Canola Oil-Degrading Bacterium *Rhodococcus*
*erythropolis* AQ5-07 from Antarctica

**DOI:** 10.3390/molecules25173878

**Published:** 2020-08-26

**Authors:** Salihu Ibrahim, Khalilah Abdul Khalil, Khadijah Nabilah Mohd Zahri, Claudio Gomez-Fuentes, Peter Convey, Azham Zulkharnain, Suriana Sabri, Siti Aisyah Alias, Gerardo González-Rocha, Siti Aqlima Ahmad

**Affiliations:** 1Department of Biochemistry, Faculty of Biotechnology and Biomolecular Sciences, Universiti Putra Malaysia, Serdang, Selangor 43400, Malaysia; ibrahimsalihu81@yahoo.com (S.I.); khadijahnabilah95@gmail.com (K.N.M.Z.); 2School of Biology, Faculty of Applied Sciences, Universiti Teknologi MARA, Shah Alam, Selangor 40450, Malaysia; khali552@uitm.edu.my; 3Department of Chemical Engineering, Universidad de Magallanes, Avda. Bulnes 01855, Punta Arenas, Región de Magallanes y Antártica Chilena, Chile; claudio.gomez@umag.cl; 4British Antarctic Survey, NERC, High Cross, Madingley Road, Cambridge CB3 0ET, UK; pcon@bas.ac.uk; 5Department of Bioscience and Engineering, College of system Engineering and Science, Shibaura Institute of Technology, 307 Fukasaku, Minuma-ku, Saitama 337-8570, Japan; azham@shibaura-it.ac.jp; 6Department of Microbiology, Faculty of Biotechnology and Biomolecular Sciences, Universiti Putra Malaysia, Serdang, Selangor 43400, Malaysia; suriana@upm.edu.my; 7National Antarctic Research Centre, B303 Level 3, Block B, IPS Building, Universiti Malaya, Kuala Lumpur 50603, Malaysia; saa@um.edu.my; 8Institute of Ocean and Earth Sciences, Universiti Malaya, B303 Level 3, Block B, Lembah Pantai, Kuala Lumpur 50603, Malaysia; 9Laboratorio de Investigacion en Agentes Antibacterianos, Facultad de Ciencias Biologicas, Universidad de Concepcion, Concepcion 4070386, Chile; ggonzal@udec.cl

**Keywords:** Antarctica, biosurfactants, canola oil, kinetics, haldane, modelling, *Rhodococcus erythropolis* AQ5-07

## Abstract

With the progressive increase in human activities in the Antarctic region, the possibility of domestic oil spillage also increases. Developing means for the removal of oils, such as canola oil, from the environment and waste “grey” water using biological approaches is therefore desirable, since the thermal process of oil degradation is expensive and ineffective. Thus, in this study an indigenous cold-adapted Antarctic soil bacterium, *Rhodococcus erythropolis* strain AQ5-07, was screened for biosurfactant production ability using the multiple approaches of blood haemolysis, surface tension, emulsification index, oil spreading, drop collapse and “MATH” assay for cellular hydrophobicity. The growth kinetics of the bacterium containing different canola oil concentration was studied. The strain showed β-haemolysis on blood agar with a high emulsification index and low surface tension value of 91.5% and 25.14 mN/m, respectively. Of the models tested, the Haldane model provided the best description of the growth kinetics, although several models were similar in performance. Parameters obtained from the modelling were the maximum specific growth rate (*q_max_*), concentration of substrate at the half maximum specific growth rate, *K_s_*% (*v/v*) and the inhibition constant *K_i_*% (*v/v*), with values of 0.142 h^−1^, 7.743% (*v/v*) and 0.399% (*v/v*), respectively. These biological coefficients are useful in predicting growth conditions for batch studies, and also relevant to “in field” bioremediation strategies where the concentration of oil might need to be diluted to non-toxic levels prior to remediation. Biosurfactants can also have application in enhanced oil recovery (EOR) under different environmental conditions.

## 1. Introduction

Many Antarctic terrestrial ecosystems are dominated by prokaryotes that play vital roles in the food chain, degradation of contaminants and biogeochemical cycles. In recent years, there has been increased appreciation of the biotechnological potential of Antarctic micro-organisms [1,2]. Antarctica is a focus of scientific research and exploration, but the negative impacts of human activity on this region can also be long lasting and have serious consequences for the continent’s native ecosystems. Antarctica is generally considered among the most pristine areas in the world. However, various areas in Antarctica have been contaminated with pollutants, including hydrocarbons [3,4]. Some Antarctic microorganisms produce biosurfactants, which increase biodegradation and the bioavailability of such contaminants. The application of bioremediation approaches in Antarctica, including oil-polluted soils, requires the use of indigenous microorganisms since, in accordance with Antarctic Treaty regulations, it is prohibited to introduce non-native biota [3,5].

Many different species of fungi, yeast and bacteria are capable of degrading hydrocarbons, with bacteria being the best-described biosurfactant producers [6,7,8]. However, details of microbial biosurfactant production remain unclear [9,10]. The bacterial genus *Rhodococcus* is well-known for its ability to produce large amounts of trehalose lipids, which can act as biosurfactants, reducing water surface tension [11,12].

The description of microbial growth kinetics is an important area in environmental biotechnology [13,14]. Models are used to explain the behaviour of microorganisms under different chemical and physical conditions, clarifying the influence of different variables including, for instance, substrate types, pH and temperature. Such models allow the prediction of substrate inhibition concentration, bacterial growth rate, substrate half saturation constant or half velocity constant and maximum substrate concentration. In order to be able to construct these models, substrate degradation and microbial growth have to be measured and modelled [15].

As happens when exposed to many contaminants, bacterial growth and the degradation of toxic substrates, such as canola oil, are characterised by a lag phase due to the need of the bacterial cells to adapt to the new environmental conditions and initiate degradation pathways. Thus, several growth phases are recognised. Specific growth rate is initially zero during the lag time (λ), after which it enters the exponential phase and reaches a maximum value (*q_max_*). Subsequently, the growth rate drops, eventually to zero, and reaches an asymptote (A) [15].

Modelling growth kinetics of canola oil-degrading bacteria is of great importance. It is often thought that vegetable oils are non-toxic, but studies have shown that lipid biodegradation products can be inhibitory to some micro-organisms, and this may affect the future development of bioremediation approaches. Studies have also shown that high oil concentrations affect the lag phase of bacterial growth, thus having an inhibitory effect on growth and degradation [16,17].

In this study, *Rhodococcus erythropolis* strain AQ5-07, a soil bacterium previously isolated from King George Island (South Shetland Islands, maritime Antarctic; [18]), was tested for its ability to degrade canola oil and produce biosurfactants, and its growth kinetics were studied and modelled. Canola oil is mostly used in research stations and is use in various cooking techniques, besides it degradation, *Rhodococcus erythropolis* strain AQ5-07 also could degrade phenol and diesel in addition to heavy metals resistance [3,18,19].

## 2. Results and Discussion

### 2.1. Growth and Biosurfactant Production

#### 2.1.1. Haemolytic Activity

Positive haemolytic activity was observed when strain AQ5-07 was screened on red blood agar, which is a widely used indicator of biosurfactant production by micro-organisms [20,21]. Similarly, Ibrahim et al. [22] reported β-haemolysis by the bacteria *B. lincheniformis*, *M. kristinae* and *S. paucimobilis.* The extent of the clear zone formed on the blood agar plate is proportional to the secretion of biosurfactant [23]. However, other confirmatory tests for biosurfactant production are required in the selection of potent biosurfactant producers [24].

#### 2.1.2. Drop Collapse Test

This is an easy and sensitive method to test for biosurfactant production [25]. In this study the *Rhodococcus erythropolis* supernatant added displaced the oil by 55 mm (Figure 1), indicating a positive drop collapse. Ibrahim et al. [22] reported a greatest oil displacement of 52 mm for *S. marcescens*, *B. lentus* by 32 mm, *M. kristinae* by 20 mm, *B. licheniformis* 18 mm and 12 mm for *B. firmus.* Therefore, the degree of drop collapse produced by this strain indicates that its biosurfactant production has the potential to improve any canola oil-degrading performance it has.

#### 2.1.3. Oil-Spreading Test

The oil-spreading approach measures the diameter of clear zones formed when a small drop of a biosurfactant-containing solution is placed into an oil layer on the surface of water [25]. There is a correlation between this and the previous drop collapse test. Biosurfactant produced by *Rhodococcus erythropolis* AQ5-07 culture gave a clear zone of 48 mm diameter in the liquid medium (Figure 1), again confirming the presence of biosurfactant in the cell-free culture broth. The oil displacement area in an oil spreading test is directly proportional to the concentration of a given biosurfactant in the solution [26], although this was not assessed quantitatively in the current study. Ibrahim et al. [22] reported that a 23 mm clear zone diameter was the lowest biosurfactant activity produced by *Bacillus licheniformis* in liquid medium for crude oil degradation, while *Micrococcus kristinae*, *Serratia marcescens* and *Pseudomonas paucimobilis* produced 51, 38 and 27 mm, respectively.

#### 2.1.4. Surface Tension Measurement

An important criterion used in the selection of biosurfactant-producing organisms is the ability to reduce surface tension to at least 40 mN/m [27]. The surface tension measurement of the cell-free supernatant here showed a considerable reduction, to 25.14 mN/m from 72.1 mN/m, clearly satisfying this criterion. Similarly, Thavasi et al. [24] reported analogous surface tension reduction in five bacterial strains that produced biosurfactants.

#### 2.1.5. Emulsification Activity

This is an indirect technique used to screen biosurfactant production. The idea is that, if the cell-free culture broth used in an assay contains biosurfactant, then it will blend the hydrocarbon present in the test solution. The study strain was able to emulsify the canola oil, which was shown by a clear emulsified layer. Emulsification activity of the cell-free broth cultured in canola oil-containing medium was measured against hexane, hexadecane, tetrahexadecane, toluene and diesel oil. The E_24_ indices were 91.5%, 88.0%, 80.7%, 52.9% and 62.0%, respectively (Figure 2). Emulsification activity in the presence of toluene and diesel was lower than for the other compounds, which may relate to the lower molecular weight of these compounds [28]. The ability of biosurfactants to emulsify water-hydrocarbon mixtures increases the degradation rate of hydrocarbons in the environment. The existence of hydrocarbon-degrading microbes in seawater allows biological degradation in this medium to be one of the most efficient techniques for contaminant removal [29]. Kumari et al. [30] reported 60.6% and 49.5% petroleum hydrocarbon degradation by biosurfactant producing *Rhodococcus* sp. NJ2 and *Pseudomonas* sp. BP10, respectively.

#### 2.1.6. Microbial Adhesion to Hydrocarbons (MATH) Assay

Rosenberg [31] established the MATH assay, which is now routinely used as a measure of microbial cell surface hydrophobicity. This assay provides an indirect method to screen for biosurfactant production, as the attachment of cells to the oil indicates the presence of surface-active compounds. Following extraction with hexadecane and tetrahexadecane, the strain showed cellular hydrophobicity of 89.6% and 83.6% to these solvents, respectively. This is consistent with Thavasi et al. [24], who reported values of 92.6% and 95.15% as the maximum adhesion to crude oil by *Lactobacillus delbrueckii* and *Pseudomonas aeruginosa*, respectively. Zhang et al. [32] similarly reported high cellular hydrophobicities of 81% in diesel for *Acinetobacter junii* VA2 and 94% for *Sphingomonas* sp. VA1. Bacterial strains with high cell hydrophobicity values are considered to be potential biosurfactant producers [33].

#### 2.1.7. Stability of Biosurfactant

The emulsification activity index values of the biosurfactant produced by *Rhodococcus erythropolis* AQ5-07 were measured at different salinity (NaCl), temperature and pH. The effects of salinities of 1–5% (*w/v*) are illustrated in Figure 3A, and the strain exhibited optimum activity at 2% salinity. The produced biosurfactant might be useful in acidic environmental conditions since it retained activity at pH 2. The emulsification index of the produced biosurfactant was inversely proportional to the salinity, but it still retained high activity at high salinity (3–4%), suggesting that the biosurfactant might be useful in marine environments. Chandran and Das [23] reported a wider salinity range for optimum emulsification activity of 2–10% in *Trichosporon asahii*. Other studies of biosurfactants produced by bacteria also show stability in presence of high salt concentration [34].

The effects of temperature on biosurfactant activity were studied across the range 5–30 °C (Figure 3B). The strain showed highest activity at 10–20 °C, with an optimum of 15 °C. These data suggest that the biosurfactants produced could be suitable for use in relatively low temperature environments and industrial systems. Most maritime Antarctic soil bacteria are psychrotolerant rather than psychrophilic [35], consistent with the scale of thermal variability typically experienced in soil habitats in the summer months in this region [36,37]. Consequently, bioremediation is usually designed and performed in the summertime when soils are unfrozen and liquid water is available. *Rhodococcus erythropolis* AQ5-07 attained maximum biosurfactant production at 15 °C, confirming that the Antarctic summer soil environment provides appropriate environmental conditions for canola oil remediation by this strain.

Figure 3C shows the effects of pH on emulsification activity over a pH range of 6–8.5. The strain showed highest activity at pH 7.5, with activity decreasing rapidly at both more alkaline and acidic pH. This could be caused by increased stability of fatty acid-surfactant micelles in the presence of NaOH and the precipitation of secondary metabolites at higher pH values. Chandran and Das [23] reported optimum pH for emulsification activity by *Trichoporon asahii* to be 2–6.

### 2.2. Optimisation of Crude Biosurfactant Production

Biosurfactant production was optimised in a batch system under facultative aerobic conditions where pH, temperature and initial canola oil substrate concentration were varied. When grown in minimal salt media at pH 7.5, the strain produced maximum biosurfactant levels when incubated at 15 °C and an initial substrate concentration of 3% (*v/v*) (Figure 4). When the incubation temperature was increased to more than 20 °C, bacterial growth and biosurfactant production were inhibited. Chandran and Das [23] reported maximum production of biosurfactant by the tropical species *Trichosporon asahii* at pH 7.5 and 35 °C. Similarly, Hamzah et al. [38] and Chen et al. [39] reported that the optimum temperature for biosurfactant production by *P. aeruginosa* UKM14T and *P. aeruginosa* S2 was 37 °C. In contrast with these strains, psychrotolerant and psychrophilic bacteria typically have growth optima below 15–20 °C.

Production of crude biosurfactant was maximum during the initial stationary phase at 72 h under all tested culture conditions. During this phase, a high concentration of crude biosurfactant was produced, which may be attributed to the discharge of cell-bound biosurfactant into the culture broth thereby increasing the extracellular biosurfactant concentration [23]. Biosurfactants are mainly produced during the early stationary phase and then remain up to the death phase. The surface tension values continue to decline until the point of critical micelle concentration where no further decrease would occur. The cause of the increase in surface tension of cultures might possibly be due to the degradation of biosurfactant in the culture media or consumption of biosurfactant for microbial survival [22].

### 2.3. Growth on Canola Oil Modelled Using Secondary Models

Bacterial growth kinetics were obtained by measuring the rate of bacterial growth over 72 h at different initial oil concentrations (Figure 5). Bacterial growth initially increased with oil concentration, reaching an optimum at initial concentration 3% (*v/v*), subsequently decreasing at concentrations higher than 3.5% (*v/v*) and being inhibited at 7% (*v/v*). The results were used to calculate the bacterial growth rate. The slope represents the growth rate, and this was plotted against the initial concentration canola oil.

The relationship between the specific growth rate (*q*) of a microbial population and the concentration of canola oil substrate (S) is a valuable tool in biotechnological applications. This relationship is described by various theoretical models [40,41]. In the current study there was an increase in cell growth rate with increasing initial canola oil concentration up 3% (*v/v*) and then a decrease at higher concentrations, indicating the inhibitory effect of canola oil. The data from the batch analyses were fitted to seven kinetics models (Figure 6).

The statistical analysis and accuracy of the kinetics models used showed that technically the best fitting model was Haldane, with the highest values for adjusted *R*^2^, *R*^2^, AF and BF nearest to unity (1.0) and the lowest value for AICc, RMSE (Table 1). However, most other models showed good and very similar fits, as illustrated in Figure 6, with the exception of Monod. The calculated values for the Haldane model constants maximum specific growth rate (*q_max_*), half saturation constant (*K_s_*) and growth inhibition constant (*K_i_*) were 0.142 h^−1^, 7.743% (*v/v*) and 0.399% (*v/v*), respectively. The equation for the Haldane model was:(1)$$\mathrm{Haldane}\;0.0669\frac{S}{S+7.743+\left(\frac{S^{2}}{0.3992}\right)}$$

The Haldane model is widely used due to its simplicity and ability to describe substrate growth inhibition kinetics by integrating substrate and growth-inhibition constants. The model is also based on a single continuous fermentation to the point of growth inhibition at high substrate concentration [42].

Most previous studies on the effects of substrate on microbial growth that have been carried out using non-toxic substrates have utilised the Monod kinetic model, whereas studies using toxic substrates such as phenol, cadmium, molybdenum, mercury, caffeine and polyethylene glycol have used one or more of the other kinetics models listed in Table 2 [41,43,44]. Data obtained in the current study demonstrate that the growth rate of this bacterium was inhibited at high canola oil concentrations.

Wayman and Tseng [45] reported that other models addressing substrate inhibition kinetics also exist, including discontinuous models. These were developed because models such as Haldane, Webb, Andrews and Noack could explain inhibitory effects on microbial growth but could not explain or adequately model certain conditions where the growth rate became zero at very high substrate concentration. Nonetheless, the major drawback of such discontinuous models remains that they cannot predict *S_m_* [46].

The use of these models in studies of the kinetics of microbial vegetable oil-degrading processes is rare because of the perceived non-toxicity of vegetable oil. However, some studies have shown that soluble lipid biodegradation products can be inhibitory, at least to certain degrading organisms, and this is pertinent to designing future bioremediation protocols [16,17]. The Haldane model has previously been used in modelling hydrocarbon degradation [47], but this is the first time that it has been utilised to model bacterial growth kinetics using canola oil as a substrate.

## 3. Materials and Methods

### 3.1. Microorganism and Shake Flask Culture Condition

Previously isolated *Rhodococcus erythropolis* AQ5-07 originally obtained from King George Island (South Shetland Islands, Antarctica) was streaked on Tween-peptone agar ((mg/L): 100 CaCl_2_, 5000 NaCl, 10,000 peptone, 18,000 agar and 5 mL/L of Tween 80) [48] and incubated at 10 °C for 72 h. Single colony was transferred to 100 mL canola oil liquid medium ((mg/L): 300 yeast extract, 1000 (NH_4_)_2_SO_4_, 200 MgSO_4_·7H_2_O, 600 KH_2_PO_4_ and 900 K_2_HPO_4_) supplemented with 3% waste canola oil (obtained from Chilean Bernardo O’Higgins Riquelme Station, northern Antarctic Peninsula, in February 2018) and grown at 10 °C and 150 rpm for 24 h [19]. Bacterial culture (10%) was then transferred to 100 mL of the canola oil medium in 250 mL Erlenmeyer flasks and incubated for 72 h at 150 rpm and 10 °C. After 72 h, 1 mL of sample was removed and centrifuged at 12,000 rpm for 10 min, after which cell biomass was measured. All experiments were done in triplicates.

### 3.2. Biosurfactant Analysis

*Rhodococcus erythropolis* AQ5-07 was screened for biosurfactant production based on haemolytic activity, drop collapse, oil spreading test, surface tension, emulsification index activity and microbial adhesion to hydrocarbon assay. In addition, the stability of the biosurfactant produced was also assessed and optimised.

#### 3.2.1. Haemolytic Activity

The haemolytic activity of *Rhodococcus erythropolis* AQ5-07 was determined by streaking the bacterium, 20 µL onto sheep blood agar plate containing 5% (*v/v*) and incubation at 15 °C for 48–72 h [49].

#### 3.2.2. Drop Collapse Test

This test was done using mineral oil as hydrocarbon substrate as described by Youssef et al. [25] with some modifications. Supernatant liquid was obtained by centrifuging bacterial culture at 12,000 rpm for 10 min. Then, a mineral oil drop (10 μL) was set on a grease-free glass slide followed by drops of the supernatant being placed onto the centre of the oil drop. Drop collapse was measured by observing the shape of the drop after 1 min. Water was used as a negative control.

#### 3.2.3. Oil-Spreading Test

The oil spreading test was conducted as explained by Rodrigues et al. [50]. Briefly, 100 μL of used engine oil was added to Petri dishes containing 50 mL dH_2_O, forming a thin layer on the surface. Then, 10 μL of cell-free culture supernatant and water (control) were carefully added to the oil surface. Following 30 s incubation, the diameter of the clear zone was measured and compared with the negative control.

#### 3.2.4. Emulsification Index Test

Emulsifying ability of the bacterium was estimated by calculating an emulsification index (E_24_) for hexadecane and diesel. To do so, 4 mL of each of hexane, hexadecane, tetrahexadecane, toluene and diesel were added to an equal volume of 4 mL of supernatant (cell-free) broth in a test tube. The resulting mixture was vortexed at high speed for 5 min and allowed to stand for 24 h. The emulsification activity index E_24_ was estimated following Shoeb et al. [49].

#### 3.2.5. Surface Tension

The surface tension was measured as described by Thavasi et al. [24]. The surface tension of the supernatant was determined using tensiometer (VCA 3000 Water surface analysis system, Billerica, MA, USA). Distilled water was used as a control.

#### 3.2.6. Microbial Adhesion to Hydrocarbon (MATH) Assay

The cellular hydrophobicity of *Rhodococcus erythropolis* AQ5-07 was measured following Zoueki et al. [51] and Rosenberg [31] with some modifications. Briefly, 24 h grown bacterial culture was collected, washed and rinsed twice with saline phosphate buffer (pH 7.4). The bacteria were then re-suspended to an absorbance (OD_600_ nm) of 1 on a spectrophotometer (U.V mini 1240 Shimadzu, Japan). Next, 300 µL of each of tetrahexadecane and hexadecane were added to 5 mL bacterial cell suspension in a clean borosilicate round-bottom glass tube. The mixture in the tube was vortexed for 2 min and set to stand for 15 min to allow tetrahexadecane and hexadecane to be separated from the aqueous phase. The aqueous phase was carefully removed, and absorbance was measured at 600 nm.

#### 3.2.7. Biosurfactant Stability Test

Obayori et al. [34] method was used to determine the biosurfactant stability of the isolate. Briefly, the cell-free broth (supernatant) was obtained by centrifuging bacterial culture at 12,000 rpm for 10 min. The pH of the collected supernatant was adjusted to 6.5, 7.0, 7.5 or 8.0, followed by the E_24_ determination. The temperature stability of the biosurfactant was also studied, measuring the E_24_ at various temperatures (5, 10, 15, 20 and 25 °C). The effects of different NaCl concentrations on biosurfactant stability were studied at concentrations of 2%, 3%, 4% and 5% (*w/v*).

#### 3.2.8. Optimisation of Biosurfactant Production

The bacterial strain was cultured at different temperatures ranging from 5–30 °C, initial substrate concentration of 0.5–5% (*v/v*) canola oil and pH ranging from 6 to 8. All experiments were carried out in triplicate. The cultures were maintained for 72–96 h at 150 rpm.

#### 3.2.9. Biosurfactant Extraction

The biosurfactant produced in the course biodegradation of canola oil was extracted using the method of Patowary et al. [52] with some little modification. For the extraction of biosurfactant, bacterial cells free supernatant was removed by centrifugation 10,000× *g* for 10 min at 4 °C. The cell free supernatant was acidified with 2 N HCl to reduce the pH to 2.0. This was then followed by extraction using chloroform and methanol (2:1 *v/v*) by vigorous shaking and allowed to stand for a few minutes until phase separation. The extracts were concentrated by removing the solvents using rotatory evaporator and then sodium sulphate anhydrous was added to remove water, producing crude biosurfactant as the resultant residue. Weight of the produced biosurfactant was evaluated and expressed in terms of milligram per millilitre (dry weight) [22].
(2)XY=(AB+CD)−XYZ
where *XY* is the dry weight of biosurfactant, *AB* is the weight of plate and *CD* is biosurfactant residue after drying, while *XYZ* is weight of the empty plate.

### 3.3. Bacterial Growth Kinetics Modelling

The bacterium was cultivated under optimum conditions for canola oil degradation in nutrient broth (to an OD_600_ of 1.3–1.4) [19] on an incubator shaker at 10 °C and 150 rpm for 84 h. About 10% (*v/v*) of the seed cultures were transferred to 100 mL of canola oil liquid medium containing initial canola oil concentrations ranging from 1%–7%. Following 12 h incubation under the same conditions, 1 mL aliquots from the culture medium were collected at 12 h intervals until 84 h and cell growth and canola oil degradation were measured [53]. The determination of the canola oil degradation kinetic variables was not possible because of the liquid nature of the oil. Therefore, only bacterial growth kinetics were studied.

Various kinetics models were used to model bacterial growth kinetics (Table 2). Kinetics modelling was carried out using bacterial growth data obtained with different initial canola oil concentrations over time. The model factors were assessed using non-linear regression and the Levenberg–Marquardt algorithm using CurveExpert Professional software (Version 2.6.5). *q*_max_ estimation was made using a steepest gradient search of the curve among the four datum points.

**Table 2 molecules-25-03878-t002:** Mathematical models developed for growth kinetics studies involving different substrate concentrations and conditions.

Model	Equation	No. of Parameters	References
Monod	$q_{max}\;\frac{S}{K_{s}+S}$	2	[54]
Haldane	$q_{max\;}\frac{S}{S+K_{s}+\left(\frac{S^{2}}{K_{i}}\right)}$	3	[55]
Teissier	$q_{max}\;\left\{1-exp\left(\frac{S}{K_{i}}\right)-exp\left(\frac{S}{K_{s}}\right)\right\}$	3	[56]
Luong	$q_{max}\frac{S}{K_{s}+S}{\left(1-\frac{S}{S_{m}}\right)}^{n}$	4	[57]
Aiba	$q_{max}\frac{S}{K_{s}+S}\;exp^{\left(\frac{-S}{K_{i}}\right)}$	4	[58]
Yano and Koga	$\frac{q_{max}S}{S+K_{s}+\left(\frac{S^{2}}{K_{i}}\right)\;\left(1+\frac{S}{K}\right)}$	4	[59]
Webb	$\frac{q_{max}S\left(1+\frac{S}{K}\right)}{S+K_{s}+\left(\frac{S^{2}}{K_{I}}\right)}$	4	[60]

Note: *q_max_*: Maximum cell growth and degradation rate (h^−1^); *K_s_*: Half saturation constant or half velocity constant (% *v/v*); *K_i_*: Inhibition constant (% *v/v*); S: Substrate concentration (% *v/v*); *S_m_*: Maximum concentration of substrate tolerated (% *v/v*); m, n, K: Curve parameters.

#### Verification of the Models

The fit of the models was verified using statistical analyses assessing the adjusted coefficient of determination (*R*^2^), root-mean-square error (RMSE), bias factor (BF), accuracy factor (AF), sum of squares errors (SSE) and corrected Akaike information criterion (AICc) [61].

## 4. Conclusions

We document that the Antarctic bacterium *Rhodococcus erythropolis* AQ5-07 is a potent biosurfactant producer and efficient canola oil degrader. The biosurfactant has high hydrophobicity, emulsification activity and low surface tension value, as well as positive drop collapse oil spreading test result. The strain was able to produce biosurfactant under optimum conditions of 2% NaCl, pH 7.5 and 15 °C. The Haldane kinetic model gave the best fit to the experimental data obtained, with lowest values of AICc, RMSE, SSE and highest *R*^2^ value, although similar overall to several other models tested. This cold-tolerant bacterium’s features make it a plausible candidate to be used for on-site bioremediation of vegetable oil-contaminated sites under low temperature.

## Figures and Tables

**Figure 1 molecules-25-03878-f001:**
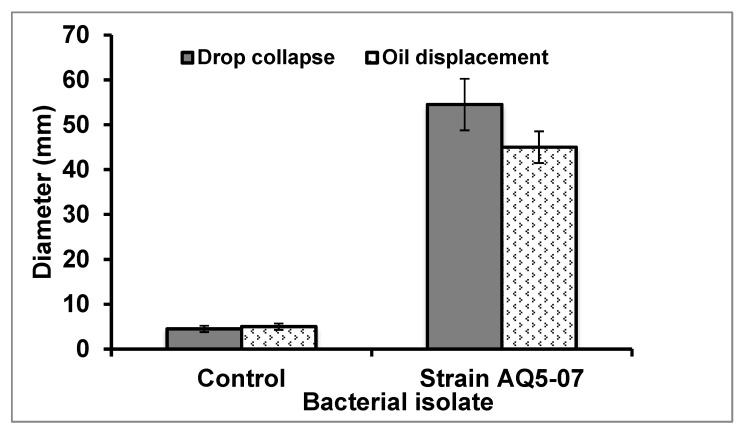
Drop collapse and oil displacement tests for supernatant fraction from *Rhodococcus erythropolis* AQ5-07 culture incubated for 72 h. Error bars represent mean ± standard deviation, *n* = 3.

**Figure 2 molecules-25-03878-f002:**
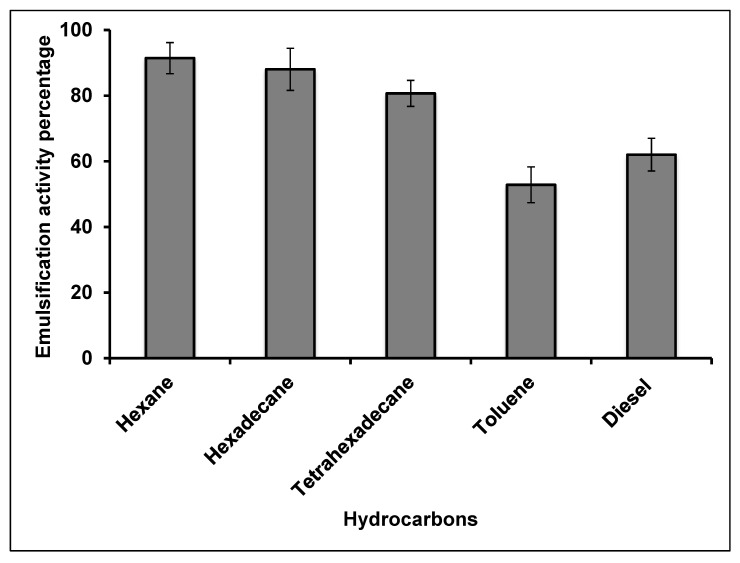
Emulsification index (E_24_) for cell-free broth from bacterial cultures after incubation for 72 h. Error bars represent mean ± standard deviation, *n* = 3.

**Figure 3 molecules-25-03878-f003:**
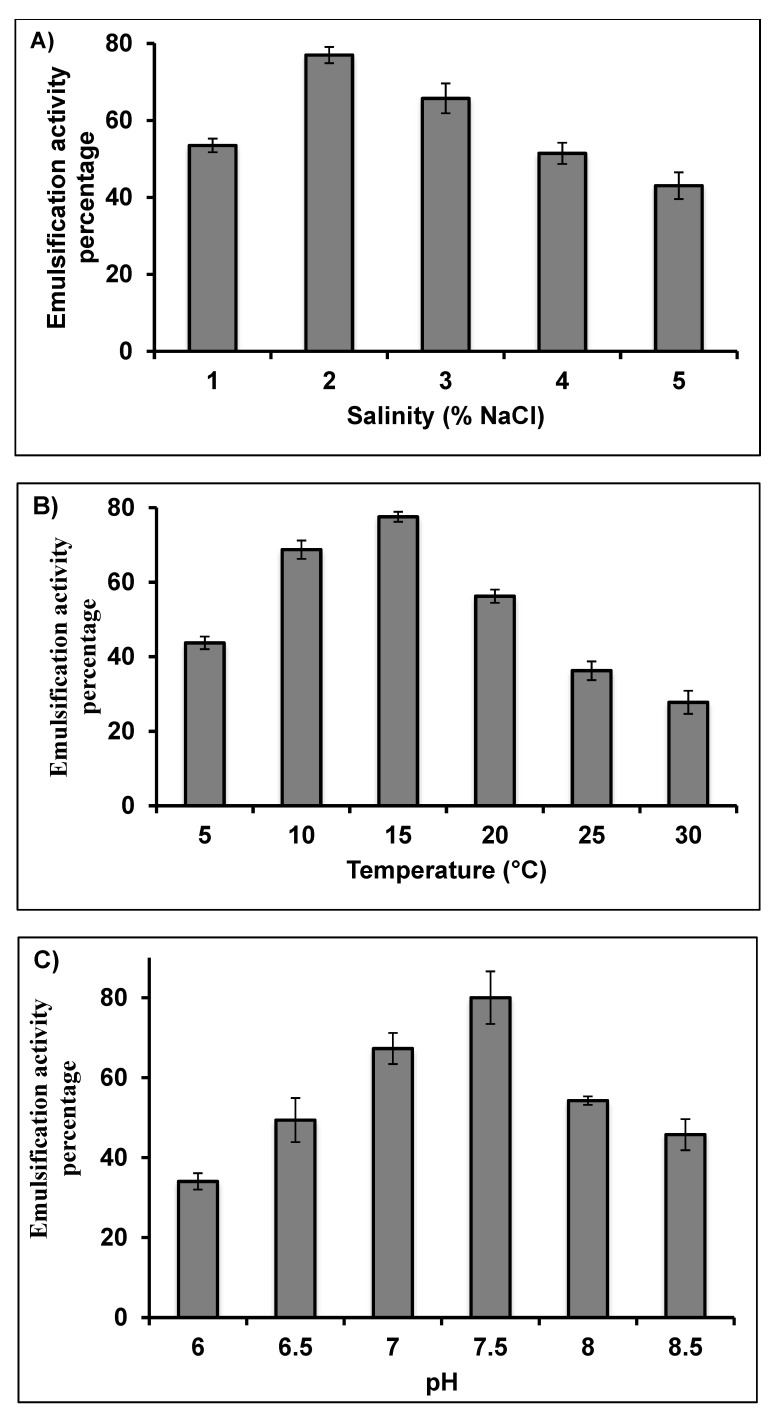
Effect of (**A**) salinity, (**B**) temperature and (**C**) pH on the emulsification activity of biosurfactant produced by *Rhodococcus erythropolis* AQ5-07. Error bars represent mean ± standard deviation, *n* = 3.

**Figure 4 molecules-25-03878-f004:**
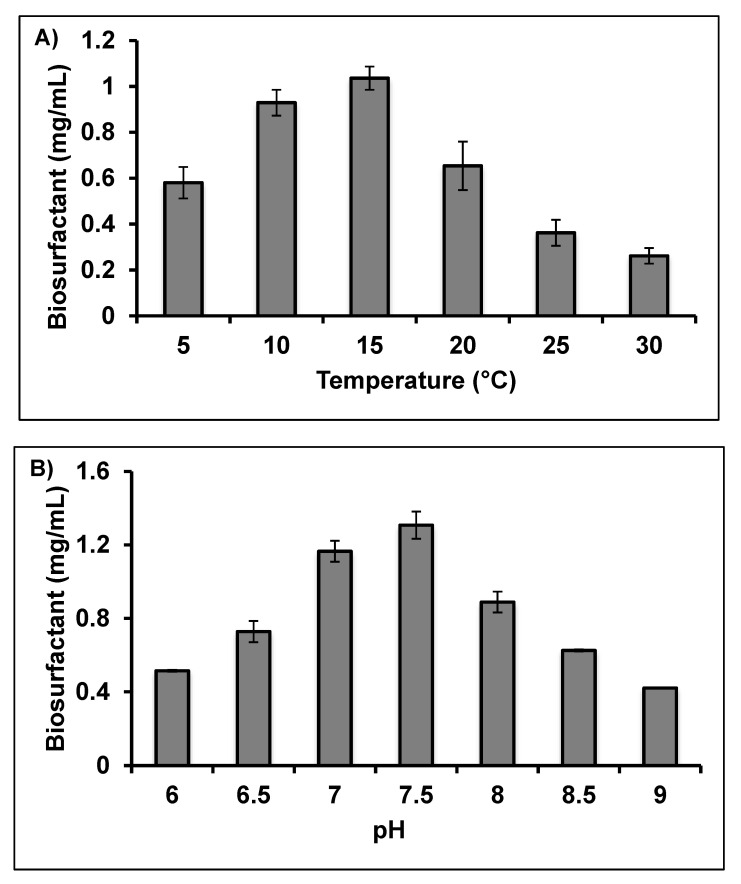
Effect of different environmental variables, (**A**) temperature, (**B**) pH and (**C**) canola oil concentrations on the biosurfactant produced by *Rhodococcus erythropolis* AQ5-07. Error bars represent mean ± standard deviation, *n* = 3.

**Figure 5 molecules-25-03878-f005:**
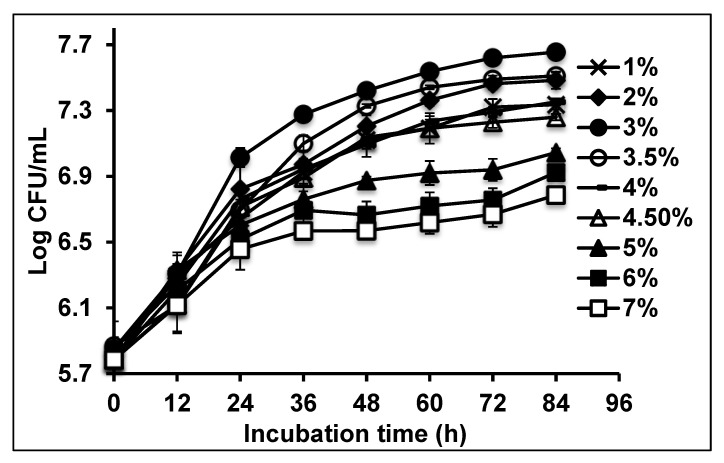
Effect of different initial canola oil concentrations on growth in *Rhodococcus erythropolis* AQ5-07. Error bars represent mean ± standard deviation, *n* = 3.

**Figure 6 molecules-25-03878-f006:**
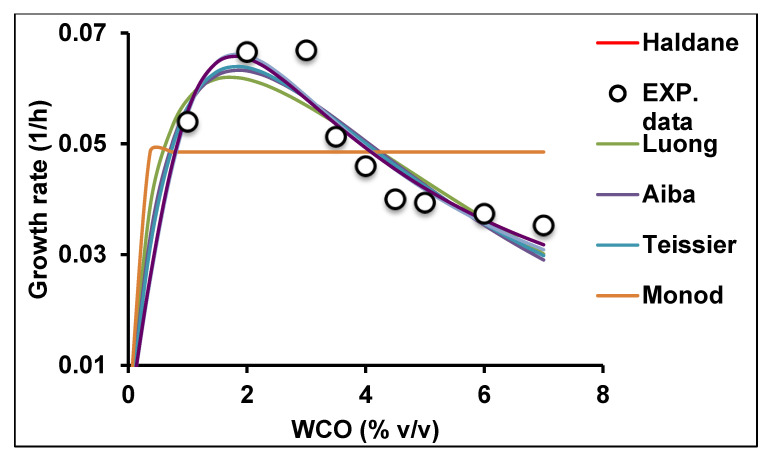
Modelling of bacterial growth kinetic experimental values using seven different kinetic models.

**Table 1 molecules-25-03878-t001:** Statistical analyses of waste canola oil growth kinetics models.

Model	Parameter	*R* ^2^	ADJ *R*^2^	AICc	RMSE	SSE	BF	AF
Haldane	3	0.9547	0.9364	−105.46	0.004733	0.000162	1.00	1.07
Teissier	3	0.8971	0.8677	−92.29	0.006949	0.000338	1.00	1.04
Monod	2	0.7181	0.8272	−90.38	0.01138	0.001166	0.98	1.12
Yano	4	0.9464	0.9196	−102.10	0.005418	0.000176	1.00	1.07
Luong	4	0.9064	0.8797	−94.68	0.006626	0.000301	1.00	1.08
Aiba	3	0.9333	0.9661	−101.57	0.005592	0.000219	1.00	1.09
Webb	4	0.9646	0.8520	−103.61	0.007349	0.000324	1.01	1.08

Note: *R*^2^: Coefficient of determination; Adj *R*^2^: Adjusted coefficient of determination; AICc: Corrected Akaike information criterion; RMSE: Root mean square error; SSE: Sum of square error; BF: Bias factor; AF: Accuracy factor.

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
