# Peer review of "Biosurfactant Production and Growth Kinetics Studies of the Waste Canola Oil-Degrading Bacterium Rhodococcus erythropolis AQ5-07 from Antarctica"

_molecules, 2020, doi:10.3390/molecules25173878_

Round 1

Reviewer 1 Report

This paper provides an interesting insight into the practicalities of bioremediation in a sensitive 'pristine' environment. Overall I found this paper clear, concise and easy to read and have only a few questions and suggestions for improvement or further explanation that would improve clarity for the reader. 

Abstract (line 39): 'bacterium containing canola oil' perhaps you mean 'bacteria in the presence of canola oil' or 'bacterium exposed to canola oil'? 

Introduction (line 90): It is not totally clear why canola oil was chosen. Perhaps a sentence to clarify if it was a model oil known to stimulate biosurfactant production in bacteria or if the oil itself is a likely contaminant in Antarctic regions.  What is the likely target contaminant in Antarctic regions? Lubricant? Ship/generator engine oil?

line 99: perhaps information on canola oil choice could be inserted here where its ability to degrade other compounds is discussed? 

Methods (line 173) why not go down to temperatures of -2 to represent minimum of unfrozen sea water?

Can Rhodococcus erythropolis strain AQ5-07 survive in seawater or is it solely a soil bacteria?

What would be the bioremediation strategy for this strain? Is this a study into natural in-situ bioremediation that may occur in the event of an oil spill or does the author propose batch growth of Rhodococcus erythropolis for biostimulation of an affected area?

Author Response

Comment 1: 

Abstract (line 39): 'bacterium containing canola oil' perhaps you mean 'bacteria in the presence of canola oil' or 'bacterium exposed to canola oil'?

Answer: Yes sir, I mean the bacterium in the presence of different canola oil concentrations. Page 2. Line 38

Comment 2:

Introduction (line 90): It is not totally clear why canola oil was chosen. Perhaps a sentence to clarify if it was a model oil known to stimulate biosurfactant production in bacteria or if the oil itself is a likely contaminant in Antarctic regions.  What is the likely target contaminant in Antarctic regions? Lubricant? Ship/generator engine oil?   Answer: The reason why canola oil was chosen is stated in line 99. Page 5   Comment 3:   line 99: perhaps information on canola oil choice could be inserted here where its ability to degrade other compounds is discussed.   Answer: Information on canola oil choice is inserted sir
Page 5. Line 98-101.   Comment 4:   Methods (line 173) why not go down to temperatures of -2 to represent minimum of unfrozen sea water?   Answer: Bioremediation in Antarctic environment is taking place summer when the soil is unfrozen and water is available. At -2oC, most of the soils were frozen. Meaning not all of it were unfrozen making it difficult for bioremediation. In addition, according to Jesus et al., 2015 who reported temperatures above 0oC are common in summer, in the King George Island region where the bacteria was isolated from. Thus, reason behind starting from 5oC. Page 8, Line 175.   Comment 5:   Can Rhodococcus erythropolis strain AQ5-07 survive in seawater or is it solely a soil bacteria?   Answer: It can also survive in seawater sir because; it can withstand 2% salt. Page 8, Line 177   Comment 6:   What would be the bioremediation strategy for this strain? Is this a study into natural in-situ bioremediation that may occur in the event of an oil spill or does the author propose batch growth of Rhodococcus erythropolis for biostimulation of an affected area?   Answer: We propose batch growth of Rhodococcus erythropolis for biostimulation of an affected area

Reviewer 2 Report

Methods for RL characterization are not sufficient. For example, you have used o surface tension, surface tension, emulsification  index, oil spreading, drop collapse and ‘MATH’ assay for cellular hydrophobicity.What chemical characteristics of the molecule?

 If the function is to remove used oils and cooking food, how would be the behavior regarding other oils used, for example, corn, sunflower, soy

What is the biosurfactant actually produced. The authors do not mention which class it belongs to.

Although it has as a novelty the use of a microorganism from Antartica, the work is not concerned with extracting and characterizing the produced metabolite, working with tests already known among the methodologies, but without making a connection with the produced molecule.

Proponents aim to use the bacteria to produce biosurfactants. they grow the same in canola oil residue and then use canola oil to produce the biosurfactant.

Why didn't they use the waste?

Only canola oil is used in the Chilean Station, and the others, what type of oil is used?

How was the Biosurfactant obtained? If the tests were carried out with the cell-free medium, it is difficult to say the conduct of the experiments because the fermented medium is complex.

Author Response

Comment 1:

If the function is to remove used oils and cooking food, how would be the behavior regarding other oils used, for example, corn, sunflower, soy.   Answer: In Antarctica, canola oil is mostly used in research stations because it contains polyunsaturated fatty acids (PUFA), low cost, low freezing point and is suitable for use in various cooking techniques. They don’t use sunflower, palm oil, or any type of oil apart from canola oil.   Comment 2:   What is the biosurfactant actually produced. The authors do not mention which class it belongs to.   Answer: The actual biosurfactant produced by Rhodococcus erythropolis is Trehalose lipids. Page 4. Line 73.   Comment 3:   Why didn't they use the waste?   Answer: The waste oil produced high amount of biosurfactant compared to pure canola oil according to our preliminary screening.   Comment 4:    Only canola oil is used in the Chilean Station, and the others, what type of oil is used?   Answer: In Chilean Station, they don’t use any type of oil apart from canola oil.

Comment 5:

How was the Biosurfactant obtained? If the tests were carried out with the cell-free medium, it is difficult to say the conduct of the experiments because the fermented medium is complex   Answer: The technique on how the biosurfactant as obtained is stated in line 186-200. Page 9

Reviewer 3 Report

Ibrahim et al. describe in their manuscript the proof of biosurfactant production by an antarctic isolate of Rhodococcus erythropolis, show the stability of emulsifying activity of the unknown compound in the culture supernatants over a range of salinities, pH, and temperatures and explore production conditions (temperature, pH).  They conclude a suitability to degrade canola oil and investigated hence growth behavior and growth inhibition with canola oil.

The manuscript is very well written and clearly presented,  and the exploration of cold-adapted biosurfactant producers is with respect to bioremediation for sure interesting, although the lack of information on the structure of the produced biosurfactant weakens the potential impact of the study, I think.

However, I experienced some major weak points in this manuscript:

How was canola oil identified as problematic environmental contaminant that has to degraded? Is there any literature about this around? Hydrocarbon contaminants refers usually more to alkanes and PAH than to triglycerids, to my knowledge. Usually, plant oils are fed on purpose as hydrophobic carbon source to biosurfactant producers to induce surfactant production. I think, from that point of view, the growth inhibition analysis is even more valuable, what may be discussed.

Figure 3 state mg biosurfactant per mL supernatant but the method section lacks a description how the surfactant was quantified/extracted?

I miss the data on 

  • biosurfactant activity at pH2 (l. 300)
  • the temperature dependent growth behavior that is mentioned in line 339 and
  • as well as the time course of biosurfactant production that is described and discussed in l. 346 and the paragraph following.

Except for the growth model section, I miss a scientific discussion of the results. Studies taken to compare physicochemical values appear somewhat random as some single papers from the wealth of biosurfactant characterizing studies and reviews. I think it would be more suitable, to include either comprehensive reviews and/or studies which are more related to this study, e.g. dealing with Rhodococcus or arctic isolates. This applies specifically for cell surface hydrophobisation (3.1.6.) because actinomycetes have by nature a cell surface structure  diverging from Lactobacillus or Gram- negatives with their LPS layer. Furthermore, I think it should be at least discussed what kind of surfactant are produced here.

Specific minor comments:

It may improve the presentation quality if the column graphics would be reduced in size. Figure 3 and 4 would fit on approximately a half page. Furthermore, I suggest to combine the proof of biosurfactant production experiments(3.1.1 till 3.1.5 or 3.1.6)  into one chapter and one figure. I appreciate that several methods are applied to avoid false-positive which may appear in single assays e.g. on blood agar, but the result of all of the methods is quite the same: R. erythropolis produces biosurfactants (and that is confirmed by all the methods).

The introduction misses references on biosurfactant production by arctic organisms and Rhodococcus/Actinomycetes. At least, the main reviews on this topic should be referenced:

Kügler et al. (2015) Surfactants tailored by the class Actinobacteria. Front Microbiol 6:212 . https://doi.org/10.3389/fmicb.2015.00212

Perfumo A, et al.,  (2018) Going Green and Cold: Biosurfactants from Low-Temperature Environments to Biotechnology Applications. Trends Biotechnol 36:277–289 . https://doi.org/https://doi.org/10.1016/j.tibtech.2017.10.016

l.98 f: As this sentence describes previous results, it may better fit in the strain description (l. 95,96): [...] maritime Antarctic [17]), that was shown to degrade phenol and diesel and to be metal resistance. In this study, we tested...

l.126: provide a reference for the blood agar applied or describe the composition (e.g. the percentage of blood)

l.184: what is the reference for optimal canola oil degradation in NB Medium?

l. 238 and all other captions: Refers n=3 to three biological replicates or three  technical replicates performed from one culture (supernatant).

l. 289. Lactobacillus does not have appeared in the manuscript to this line. The full name should be stated.

l.291 Typo in Acinetobacter

l.337 what is MSM medium? this term appears here for the first time; missing comma after "7.5"

l.437 The authors did not show efficient canola oil degradation

l.438 Hydrophicity of the biosurfactant is not investigated, is it?

l.439. What is the meaning of the term "The isolate is stable" ?

l. 544 Typo : HPTLC

Author Response

Comment 1

Figure 3 state mg biosurfactant per mL supernatant but the method section lacks a description how the surfactant was quantified/extracted?   Answer: Description on how it was extracted is included in 2.2.9. Page 9   Comment 2  

  • biosurfactant activity at pH2 (l. 300)
  • the temperature dependent growth behavior that is mentioned in line 339 and

as well as the time course of biosurfactant production that is described and discussed in l. 346 and the paragraph following   Answer: We mean that when the temperature is increased to more than 20 °C, the bacterial growth is slow thereby leading to less production of the biosurfactant and at the long ran inhibiting the bacteria to produce biosurfactant. At 15 - 20 °C the bacteria produce optimum biosurfactant. Thank you. Page 19, Line 358-364.   Comment 3:   It may improve the presentation quality if the column graphics would be reduced in size. Figure 3 and 4 would fit on approximately a half page. Furthermore, I suggest to combine the proof of biosurfactant production experiments (3.1.1 till 3.1.5 or 3.1.6) into one chapter and one figure. I appreciate that several methods are applied to avoid false-positive which may appear in single assays e.g. on blood agar, but the result of all of the methods is quite the same: R. erythropolis produces biosurfactants (and that is confirmed by all the methods).   Answer: All these chapters are independent chapters. So, it will look somehow to merge all the chapters as it leads to misunderstanding by the readers.   Comment 4: The introduction misses references on biosurfactant production by arctic organisms and Rhodococcus/Actinomycetes. At least, the main reviews on this topic should be referenced: Kügler et al. (2015) and Perfumo A, et al.,  (2018)   Answer: Duly referenced sir. References no. 11 and 12.    Comment 5:   l.98 f: As this sentence describes previous results, it may better fit in the strain description (l. 95,96): [...] maritime Antarctic [17]), that was shown to degrade phenol and diesel and to be metal resistance. In this study, we tested...   Answer: Noted sir.   Comment 6:   l.126: provide a reference for the blood agar applied or describe the composition (e.g. the percentage of blood)   Answer: Reference and description provided sir. Page 6, 127-128.   Comment 7:   what is the reference for optimal canola oil degradation in NB Medium?   Answer: Reference provided sir. Page 9, reference no. 19.   Comment 8:   and all other captions: Refers n=3 to three biological replicates or three  technical replicates performed from one culture (supernatant)   Answer: It refers to biological replicate. Meaning each experiment was conducted in triplicate   Comment 9:   289. Lactobacillus does not have appeared in the manuscript to this line. The full name should be stated.   Answer: The full name was stated. Page 15, 309.   Comment 10:   Typo in Acinetobacter   Answer: Corrected sir. Page 15, Line 331.   Comment 11:    what is MSM medium? this term appears here for the first time; missing comma after "7.5"   Answer: The full meaning of MSM is stated.   Comment 12:   The authors did not show efficient canola oil degradation   Answer: The efficient canola oil degradation are been reported in Ibrahim et al 2020 and duly cited as No [19] on the list Another one was accepted in Electronic Journal of Biotechnology as (Ibrahim et al 2020b), waiting to be published.    Comment 13:   What is the meaning of the term "The isolate is stable" ?   Answer: The sentence is paraphrased. Page 25, Line 458   Comment 14:   Typo : HPTLC   Answer: Corrected sir. Page 30, Line. Reference No. 20    

Reviewer 4 Report

Removal of oils from the environment and waste ‘grey’ water using biological approaches is a hot and important subject . However, procedures to do so are usually very expensive and sometimes not completely effective. The authors proposed the study of an indicator soil bacterium cold-adapted which is the Antarctic  Rhodococcus erythropolis strain AQ5-07. The bacterium was screened for biosurfactant production ability. The growth kinetics of the bacterium containing canola oil was studied and tested by the use of different models. The Haldane model showed the best description of the growth kinetics, although several models were similar in performance. These models help to establish the best ‘in field’ bioremediation strategy for environmental protection.

It is a well documented and written paper. It should be of high interest to the scientific community , specifically for those working in the area of environmental pollution.Data are evaluated in an appropriate way .

The researchers team seems to be qualified and experienced on this scientific domain .

The paper could be published in its present as it is well written , methodology is well explained in order to be reproducible and bibliography is up to date .

Author Response

Comment 1

The reviewer only commend us by saying

The researchers team seems to be qualified and experienced on this scientific domain.

The paper could be published in its present as it is well written, methodology is well explained in order to be reproducible and bibliography is up to date.   Answer: Thank you very much.

Round 2

Reviewer 2 Report

The authors talk about the biosurfactant produced as lipids trehalose and cites two references that are reviews. I return to affirm that it would be interesting to have characterized this biosurfactant, as it is an interesting work with a microorganism that lives at low temperature and this fact leads us to think that a mass spectrometry could better characterize and bring news in terms of new congeners, because it is known that among trehalose lipids there are different molecules as mentioned in reference 11.

How pure is the biosurfactant obtained?

Author Response

Answer:

Thank you very much sir, normally biosurfactants are characterised using mass spectrometry. The biosurfactant produced by Rhodococcus erythropolis is trehalose dicorynemycolate (Trehalose 6,6′-dimycolate) as a sole non-reducing sugar according to many references such as Franzetti et al., 2010; Peng et al 2007; Pacheco et al 2010; Rapp et al 1979 to mention few among other references. Whole genome analysis of strain AQ5-07 using PacBio RSII platform shows that this strain is similar to Rhodococcus erythropolis with an ANI value of 98.70.

References:

Franzetti, A., Gandolfi, I., Bestetti, G., Smyth, T. J. P., and Banat, I. M. (2010). Production and applications of trehalose lipid biosurfactants. European Journal of Lipid Science and Technology, 112, 617–627.

Pacheco, G. J., Ciapina, E. M. P., de Barros Gomes, E., and Pereira Junior, N. (2010). Biosurfactant production by Rhodococcus erythropolis and its application to oil removal. Brazilian Journal of Microbiology, 41(3), 685–693. https://doi.org/10.1590/S1517-83822010000300019

Peng, F., Liu, L., and Shoa, Z. (2007). An oil-degrading bacterium: Rhodococcus erythropolis strain 3C-0 and its biosurfactants. Journal of Applied Microbiology, 102, 1603–1611. https://doi.org/http://dx.doi.org/10.1016/j.cej.2015.11.026

Rapp, P., Bock, H., Wray, V., and Wagner, F. (1979). Formation, isolation and characterization of trehalose dimycolates from Rhodococcus erythropolis grown on n-alkanes. Journal of General Microbiology, 115(2), 491–503. https://doi.org/10.1099/00221287-115-2-491

Reviewer 3 Report

I see the scientific questions properly addressed.

Author Response

Thank you very much.